# Electron-momentum dependence of electron-phonon coupling underlies dramatic phonon renormalization in YNi$_2$B$_2$C

Philipp Kurzhals[1], Geoffroy Kremer[2], Thomas Jaouen[2,3], Christopher W. Nicholson [2], Rolf Heid [1], Peter Nagel[1], John-Paul Castellan[1,4], Alexandre Ivanov [5], Matthias Muntwiler [6], Maxime Rumo [2], Bjoern Salzmann [2], Vladimir N. Strocov [6], Dmitry Reznik [7,8], Claude Monney [2] & Frank Weber [1✉]

Electron-phonon coupling, i.e., the scattering of lattice vibrations by electrons and vice versa, is ubiquitous in solids and can lead to emergent ground states such as superconductivity and charge-density wave order. A broad spectral phonon line shape is often interpreted as a marker of strong electron-phonon coupling associated with Fermi surface nesting, i.e., parallel sections of the Fermi surface connected by the phonon momentum. Alternatively broad phonons are known to arise from strong atomic lattice anharmonicity. Here, we show that strong phonon broadening can occur in the absence of both Fermi surface nesting and lattice anharmonicity, if electron-phonon coupling is strongly enhanced for specific values of electron-momentum, **k**. We use inelastic neutron scattering, soft x-ray angle-resolved photoemission spectroscopy measurements and ab-initio lattice dynamical and electronic band structure calculations to demonstrate this scenario in the highly anisotropic tetragonal electron-phonon superconductor YNi$_2$B$_2$C. This new scenario likely applies to a wide range of compounds.

[1] Institute for Quantum Materials and Technologies, Karlsruhe Institute of Technology, 76021 Karlsruhe, Germany. [2] Département de Physique and Fribourg Center for Nanomaterials, Université de Fribourg, 1700 Fribourg, Switzerland. [3] Univ Rennes, CNRS, IPR (Institut de Physique de Rennes) - UMR 6251, F-35000 Rennes, France. [4] Laboratoire Léon Brillouin (CEA-CNRS), CEA Saclay, F-91911 Gif-sur-Yvette, France. [5] Institut Laue-Langevin, 71 avenue des Martyrs CS 20156, 38042 Grenoble Cedex 9, France. [6] Paul Scherrer Institut, Swiss Light Source, 5232 Villigen PSI, Switzerland. [7] Department of Physics, University of Colorado at Boulder, Boulder, CO 80309, USA. [8] Center for Experiments on Quantum Materials, University of Colorado at Boulder, Boulder, CO 80309, USA. ✉email: frank.weber@kit.edu

Interacting degrees of freedom in solids underlie new emergent ground states and competing phases with potential for new functionalities. Vibrations of the atomic lattice, i.e. phonons, can couple to electrons[1], magnetic[2], or orbital degrees of freedom[3]. In particular, electron–phonon coupling (EPC) received a lot of attention as a microscopically understood origin of superconductivity. Furthermore, EPC has recently been in the focus of investigations of materials with competing phases, such as cuprates[4–8] and layered transition-metal dichalcogenides[9–13].

Reduced phonon lifetimes are typically related to nesting where large sections of the Fermi surface (FS) are connected by a single phonon wave vector[14–16]. The hallmark of nesting is a singularity or at least a strong peak in the electronic susceptibility for a particular phonon wave vector **q** connecting parallel sections of the FS[17,18].

Such phonon anomalies received a lot of attention recently in the context of charge density wave (CDW) formation, which is often driven by a soft phonon mode triggering a structural distortion at the CDW transition. In some CDW compounds, such as 1D conductors, this behavior is indeed related to the nesting of the FS[19–21]. Yet in others, such as $2H$-NbSe$_2$, nesting is absent[22–26] and cannot explain the soft-mode properties. While the CDW in $2H$-NbSe$_2$ originates from EPC[22,27], it has been proposed that the phonon softening and broadening on cooling towards the CDW transition temperature $T_{CDW}$ can be explained only by taking into account lattice fluctuations[25], and anharmonic effects may play an important role[24]. Phonon anomalies in cuprates remain enigmatic though a strong response to the onset of superconductivity is evident[4,7]. In other cases, an expected phonon broadening related to nesting is absent[28] or incomplete[29]. In principle, the phonon momentum, **q**, dependence of the EPC itself, expressed in the EPC matrix elements $g_{\vec{k}+\vec{q},\vec{k}}^{\vec{q}\lambda}$, can determine the wavevector of the phonon broadening in the absence of FS nesting[26,27]. Yet, changes of the phonon linewidth because of the temperature dependence of $g_{\vec{k}+\vec{q},\vec{k}}^{\vec{q}\lambda}$ up to room temperature are expected to be small.

Here, we propose a scenario in which the electron momentum, **k**, dependence of the EPC matrix elements comes into play. In such a scenario, the $g_{\vec{k}+\vec{q},\vec{k}}^{\vec{q}\lambda}$, is particularly large for electrons on certain parts of the FS. We show that such **k**-selective EPC can be strong enough to significantly broaden phonons even when **k**-integrated quantities like the electronic susceptibility $\chi_q$ lack particular features at the phonon momentum **q**. In this case, the broadening can sensitively depend on the temperature-induced changes of electronic states at the Fermi energy. This scenario can explain large temperature-dependent phonon line widths in the absence of both nesting and anharmonicity.

We demonstrate this scenario on the electron–phonon superconductor YNi$_2$B$_2$C ($T_c$ = 15.2 K). YNi$_2$B$_2$C is known for unusual low-temperature phonon lineshapes reflecting the anisotropic superconducting energy gap[30–32], EPC distributed over 160 meV in phonon energy[33] and phonons with unusual eigenvectors mediating superconductivity[34]. This work focuses on the microscopic origin of strongly increased line widths of certain phonons observed over a large range of wavevectors in the normal state at low temperatures, i.e. $T$ = 20 K.

We first argue that previously proposed explanations for the phonon properties in YNi$_2$B$_2$C, i.e., FS nesting[35], can be ruled out. Then we demonstrate that results of our comprehensive inelastic neutron scattering (INS) and soft x-ray angle-resolved photoemission spectroscopy (SX-ARPES) measurements agree well with ab-initio lattice dynamical and electronic band structure calculations, which allows us to use these calculations to gain insights into the microscopic origin of EPC. The calculations highlight the

importance of 2D electronic joint density of states (2D-eJDOS). It is defined as the usual electronic joint density of states (eJDOS) but evaluated for 2D slices of the reciprocal space at a specific component of $\mathbf{q}k_z$, where the z-direction is defined to be parallel to the crystallographic c-axis (Fig. 1a, b). The results highlight the decisive role of the interplay between the **k**-dependence of the EPC combined with a strongly **k**-dependent 2D-eJDOS, and explain strongly temperature-dependent phonon broadening in the absence of FS nesting and lattice anharmonicity.

Electron–phonon coupling renormalizes the phonon energy and reduces the phonon lifetime through emission or absorption of electron–hole pairs. Quantitatively, phonon energy is affected by the real part of the **q** dependent electronic susceptibility $\chi_q$, whereas the lifetime is determined by the imaginary part (see Feynman diagram in Fig. 1a). The former depends on electronic states near as well as far from the Fermi surface, whereas only electronic states near the FS contribute to the latter.

Thus knowledge of both electrons and phonons as well as the coupling between them is necessary to understand phonon self-energy in metals. Electronic states near the FS can be accurately measured by angle-resolved photoemission (ARPES) and the phonon linewidths can be measured by neutron scattering. On

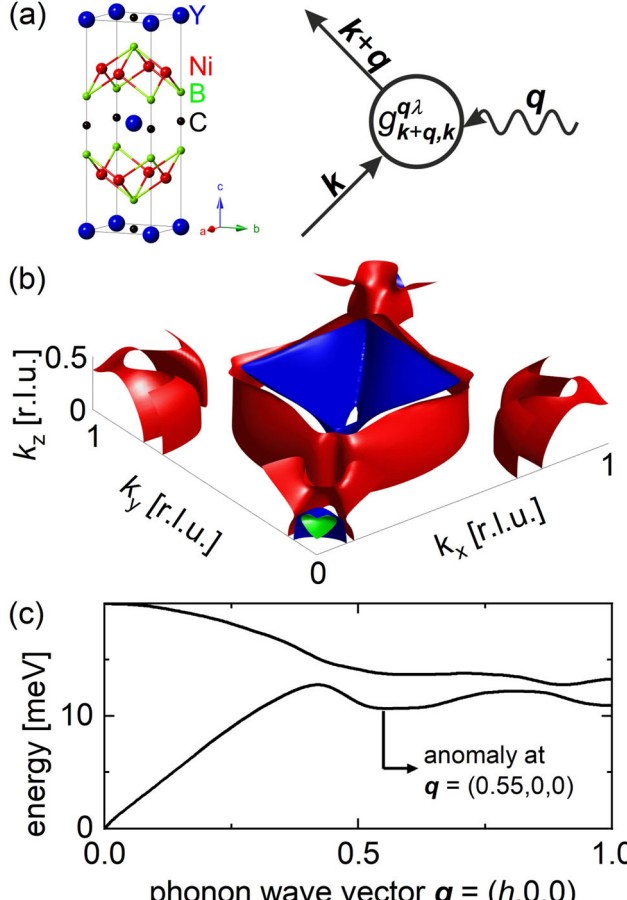

**Fig. 1 Electron–phonon coupling. a** Crystal structure of YNi$_2$B$_2$C ($I4/mmm$, $a = b = 3.51$Å, $c = 10.53$Å) and Feynman diagram of the EPC process including the EPC matrix element $g_{\mathbf{k}+\mathbf{q},\mathbf{k}}^{\mathbf{q}\lambda}$. Wave vectors **k** and **q** refer to electron and phonon momentum space, respectively. **b** Calculated partial FS of YNi$_2$B$_2$C. Colors denote different bands crossing the Fermi energy. **c** Calculated phonon dispersion featuring an anomaly at $\mathbf{q} = (0.55, 0, 0)$. Note nearly parallel sections of the Fermi surface at $k_z = 0.5$ where 2D-eJDOS is enhanced.

the other hand, measurements of electronic states far from the FS that need to be included to obtain phonon peak shift due to interaction with electrons are a lot more challenging especially for the states that are unoccupied. Thus it is easier to compare theory with experiments for the phonon lifetimes than for the peak positions.

Phonon lifetimes are reduced when phonons are absorbed or emitted by electronic excitations across the Fermi surface. Since the energy and momentum are conserved, only occupied electronic states in a small energy window $E_F - E_{phon}^{q\lambda}$ can contribute to this process, where $E_{phon}^{q\lambda}$ is the energy of a phonon mode with wave vector $\mathbf{q}$ in the dispersion branch $\lambda$ (Fig. 1b). Reduced lifetimes translate into large phonon spectral line widths.

Two quantities are relevant for determining phonon line broadening due to the EPC: (1) The likelihood that a particular phonon with $E_{phon}^{q\lambda}$ is scattered/absorbed by a particular electron in a state with wave vector $\mathbf{k}$ and energy $E_{el}^{k}$, which is expressed as the EPC matrix element $g_{\vec{k}+\vec{q},\vec{k}}^{\vec{q}\lambda}$. (2) The imaginary part of the electronic susceptibility $\chi_q''$ reflects the number of electronic states at the FS which are connected by $\mathbf{q}$, i.e. the eJDOS, which is defined as $\sum_{\vec{k}} \delta(E_{\vec{k}} - E_F)\delta(E_{\vec{k}+\vec{q}} - E_F)$ and equivalent to the so-called nesting function $\xi(\mathbf{q})$[36]. (In the following, we will use the term eJDOS. The equivalent term *nesting function* could cause confusion since conventional FS nesting plays absolutely no role in the reported effects.) Both $g_{\vec{k}+\vec{q},\vec{k}}^{\vec{q}\lambda}$ and $\chi_q$ are in general $\mathbf{q}$ dependent and, therefore, their interplay in $\mathbf{q}$-space determines the $\mathbf{q}$-dependence of phonon linewidths (Fig. 1c). Eq. (1) describes the contribution $\gamma_{EPC}^q$ to the phonon linewidth $\Gamma \propto$ (phonon lifetime)$^{-1}$ because of EPC:

$$\gamma_{EPC}^q = \pi\omega_{q\lambda} \sum_{\vec{k}} \left| g_{\vec{k}+\vec{q},\vec{k}}^{q\lambda} \right|^2 \delta(\epsilon_{\vec{k}} - \epsilon_F)\delta(\epsilon_{\vec{k}+\vec{q}} - \epsilon_F) \qquad (1)$$

Apart from their intrinsic $\mathbf{q}$ dependence, the EPC matrix elements $g_{\vec{k}+\vec{q},\vec{k}}^{\vec{q}\lambda}$ can amplify or suppress contributions to the phonon line width for electronic states at different $\mathbf{k}$. This $\mathbf{k}$-selectivity has not yet been considered in the analysis of EPC. We show that it can have a profound impact on the lattice dynamical properties as exemplified by the behavior of the phonon broadening in YNi$_2$B$_2$C.

## Results

**Lattice dynamics—previous results.** YNi$_2$B$_2$C offers unique insights into the interplay between electronic and lattice degrees of freedom via superconductivity-induced phonon anomalies[30,32,37]. Phase competition of superconductivity with CDW order[35], the role of FS nesting[38–40] as well as its lattice dynamics have been investigated extensively[30,41] including in our earlier work[32–34]. We first demonstrate that temperature and $\mathbf{q}$ dependence of phonon renormalization in YNi$_2$B$_2$C is inconsistent with either standard mechanism: FS nesting or anharmonicity.

Overall, ab-initio lattice dynamical calculations of YNi$_2$B$_2$C ($I4/mmm$, $a = b = 3.51$ Å, $c = 10.53$ Å) agree with phonon spectroscopy results at energies up to 160 meV with regard to both phonon energies and corresponding phonon line widths due to the EPC[32–34]. The strongest phonon anomalies are observed in $c$-axis polarized transverse acoustic (TA) phonon modes[34]. One such mode at $\mathbf{q} = (0.5, 0.5, 0)$, the $M$ point of the Brillouin zone (BZ), is measured at $\mathbf{Q} = (0.5, 0.5, 7)$, i.e., in the BZ adjacent to $\tau = (1, 0, 7)$ (Fig. 2a). Another strong coupling TA mode is observed at $\mathbf{q} = (0.55, 0, 0)$, i.e., about halfway between the $\Gamma$ and $X = (1, 0, 0)$ points of the BZ. For simplicity, we call it here the $X/2$ anomaly. We measured it at $\mathbf{Q} = (0.45, 0, 7)$, i.e. close to the

same BZ center as the $M$ point phonon, i.e. $\tau = (1, 0, 7)$ (Fig. 2b). The $X/2$ anomaly is the only part of the phonon dispersion, which is not very well captured by ab-initio calculations in that the observed anomaly is stronger than predicted[34].

Key enigmatic results are (1) that strong phonon broadening appears over a wide range of phonon wave vectors $\mathbf{q} = (h, 0, 0)$, $0.4 \leq h \leq 0.75$, i.e. $\mathbf{q} \approx X/2$, but quickly vanishes going away from $\mathbf{q} = (0.5, 0.5, 0)$ along the [110] direction and (2) that broadening of the TA mode at $X/2$ is much stronger than at the $M$ point. Both TA phonons display a pronounced broadening upon cooling from room temperature to $T = 20$ K easily identified in high-energy resolution INS data (see the "Methods" section) (Fig. 2a–d). The new data allows for a detailed study of the momentum dependence of phonon broadening (see the "Results" section and Figs. 5, S7–S9).

The observed broadening at $\mathbf{q} \approx X/2$ is much stronger than at the $M$ point (Fig. 2a, b). In contrast, the calculated eJDOS[34] along the [100] and [110] directions displays a peak only for $\mathbf{q} = (0.5, 0.5, 0)$ (Fig. 2e). Yet, the calculations correctly predict the broad momentum range along the [100] direction over which phonon broadening is observed[34]. Therefore, nesting is unlikely to be the origin of the observed strong phonon renormalization. On the other hand, EPC matrix elements $g_{\vec{k}+\vec{q},\vec{k}}^{q\lambda}$ [see Eq. (1)] can underlie a $\mathbf{q}$ dependence not reflected by the FS geometry. However, the matrix elements are expected to change only for temperatures on the order of an *electron-volt* corresponding to much higher values than those in Fig. 2.

Anharmonic broadening of phonons is typically attributed to phonon–phonon scattering which is typically strong at elevated temperatures.[42,43] At low temperatures anharmonic broadening can occur near or at structural phase transitions as it is argued for the case of the CDW transition in $2H$-NbSe$_2$ ($T_{CDW} = 33$ K).[24] In fact, it was argued that YNi$_2$B$_2$C is close to a structural phase transition related to the soft mode at $\mathbf{q} \approx (0.5, 0, 0)$ only precluded by the onset of superconductivity.[35] Hence, the large phonon linewidth just above $T_c = 15.2$ K could still be anharmonic. However, previously reported superconductivity-induced changes of the line shape of the anomalous TA phonons to rule out a non-electronic origin.[30] This redistribution of phonon spectral weight[32] directly reflects the opening of the superconducting gap $2\Delta$ (for more details see Fig. S1 and Supplementary Note 1). A model[37] that accurately describes these observations is purely EPC-based, which is possible only if phonon broadening in YNi$_2$B$_2$C occurs only because of electronic scattering, i.e., is EPC in nature.

In the following, we present evidence that the strong TA phonon broadening in YNi$_2$B$_2$C originates from EPC that is strongly enhanced on parts of the Fermi surface, which are connected by the correct values of the phonon momentum. This scenario explains the results from phonon spectroscopy (see Fig. 2a, b) in the absence of FS nesting or lattice anharmonicity.

**Band structure from electron spectroscopy.** There has been considerable discussion about FS nesting for $\mathbf{q} \approx X/2$ in (Lu/Y) Ni$_2$B$_2$C including theoretical[35,38,44] and experimental work[39,40,45,46]. Yet, the clearest signature of electron–phonon coupling, the $T_c$-induced phonon effect, appears over a large momentum region along the [100] direction[32], which is inconsistent with nesting. In order to clarify the origin of this EPC, we performed SX-ARPES measurements on samples cut from the large YNi$_2$B$_2$C single crystal, which we used for our current and also previous INS studies[32–34].

SX-ARPES measurements at photon energies $h\nu \geq 650$ eV have a resolution, which is sharp enough to reliably resolve the $k_z$-dispersion effects in YNi$_2$B$_2$C. (The half out-of-plane periodicity of the BZ, i.e. the $\Gamma - Z$ distance [see Fig. 3f], is $2\pi/c =$

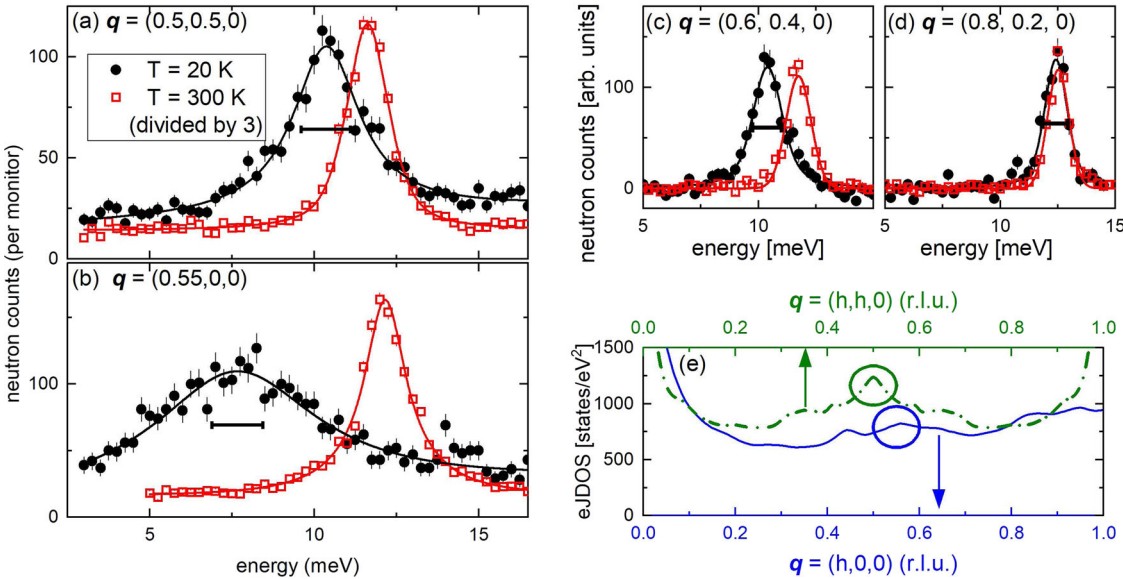

**Fig. 2 Phonon broadening in YNi$_2$B$_2$C. a**, **b** Raw data from inelastic neutron scattering (INS) revealing the strong softening and broadening of $c$-axis polarized acoustic phonons at **a** $\mathbf{Q} = (0.5, 0.5, 7)$, **b** $\mathbf{Q} = (0.45, 0, 7)$, **c** $\mathbf{Q} = (0.4, 0.4, 7)$ and **d** $\mathbf{Q} = (0.2, 0.2, 7)$ corresponding to the reduced phonon wave vectors $\mathbf{q} = (0.5, 0.5, 0)$, $\mathbf{q} = (0.55, 0, 0)$, $\mathbf{q} = (0.6, 0.4, 0)$ and $\mathbf{q} = (0.8, 0.2, 0)$, respectively, within the Brillouin zone adjacent to $\boldsymbol{\tau} = (1, 0, 7)$. Horizontal bars indicate the full-width at half-maximum of the peak observed at $T = 300$ K. Error bars represent s.d. **e** Calculated total electronic joint density-of-states (eJDOS) along $\Gamma - X$ (solid blue line, bottom axis) and $\Gamma - M - \Gamma$ (dashed green line, top axis) in $\mathbf{q}$ space. Circles indicate the phonon wave vectors corresponding to those shown in (**a**) and (**b**).

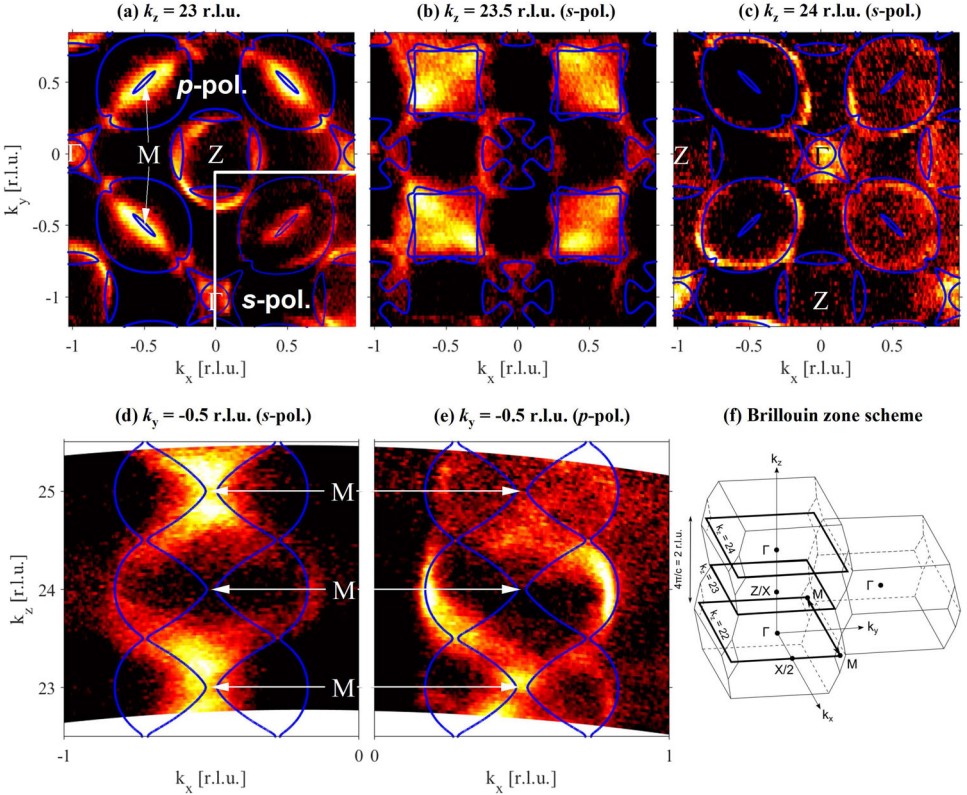

**Fig. 3 Fermi surface from soft x-ray angle-resolved photoemission spectroscopy.** Comparison of calculated Fermi surface (lines) and SX-ARPES intensities (white/black: high/low intensities) observed in the $\Gamma - X/Z - M$ plane of the Brillouin zone (BZ) with (**a**) $k_z = 23$ r.l.u. $= 13.7$ Å$^{-1}$ achieved at $hv = 693$ eV, (**b**) 23.5 r.l.u. $= 14$ Å$^{-1}$ at 725 eV and (**c**) 24 r.l.u. $= 14.3$ Å$^{-1}$ at 758 eV. **d**, **e** Calculated Fermi surface (lines) and SX-ARPES intensities for the $k_x - k_z$ plane at $k_y = -0.5$ r.l.u. Measurements shown in **a–d** were performed with $p$ polarization except for the lower right quadrant in (**a**) which was obtained with $s$ polarized light as data in (**e**). **f** Sketch of the BZ of YNi$_2$B$_2$C. Note that $X$ and $Z$ points are equivalent. Wave vectors are given in reciprocal lattice units (r.l.u.) of $\left(2\pi/a, 2\pi/b, 2\pi/c\right)$.

0.597 Å$^{-1}$ [$k_z = 1$ r.l.u.], which corresponds to a difference in the incident photon energy of $\Delta(h\nu) \approx 65$ eV). We compared the experimentally observed band structure with density functional theory (DFT) calculations carried out in the framework of the mixed basis pseudopotential method[47] using the local density approximation (LDA) (for details see ref. [34]).

Selected cuts of the FS in the $k_x - k_y$ plane in Fig. 3a–c and the $k_x - k_z$ plane in Fig. 3d, e overlaid with corresponding DFT calculations (solid blue lines), show a good agreement (see also Figs. S2 and S3 in SI). We focus on the vicinity of the $M$ point, i.e. at $\mathbf{k} = (0.5, 0.5, k_z)$, since the DFT predicts a square-shaped FS around it for certain $k_z$ values prone to nesting[34]. Our results demonstrate the significant evolution of the FS at $M$: An elliptical FS stretches between the four visible $\Gamma$ points at $k_z = 23$ r.l.u. (Fig. 3a, $h\nu = 693$ eV), a nearly square-shaped feature is found at $k_z = 23.5$ r.l.u. (Fig. 3b, $h\nu = 725$ eV), and a large nearly circular ellipse is observed at $k_z = 24$ r.l.u. (Fig. 3c, $h\nu = 758$ eV) with the long axis rotated by 90° relative to the orientation at $k_z = 23$ r.l.u.. Note that due to the particular symmetry of the BZ, the positions of $\Gamma$ and $Z$ points are reversed upon moving from $k_z = 23$ r.l.u. to $k_z = 24$ r.l.u. (see Fig. 3f).

The experimentally observed spectral weight with a square-like shape for $k_z = 23.5$ r.l.u. is relatively diffuse, in contrast to the sharp FS expected from the calculated Fermi contour (Fig. 3b). This originates from two features of the band structure in the vicinity of the $M$ point: (1) The $k_z$ dispersion of the inner band is very strong (Figs. 3d, e and S4c) and (2) both the inner and outer bands forming the square have a shallow in-plane energy dispersion of <0.5 eV near $E_F$ (see Fig. S3c). Hence, due to the uncertainty of the inner potential $V_0$ used in rendering photon energy into $k_z$ and to the limited $k_z$ and energy resolution of the experiment, we pick up intensity inside the square.

We used both linear vertical, i.e. $p$, and linear horizontal, i.e. $s$ polarization of the light in order to exploit photoemission matrix element effects. $p$ polarization is sensitive to the inner (outer) band for odd (even) values of $k_z$, whereas $s$ polarization reveals inner and outer bands for both odd and even $k_z$. This is evident in Fig. 3a, c where the spectral weight is located at the center and at the edges of the ellipse at the $M$ point, respectively, with $p$, but uniformly distributed with $s$ (bottom right in Fig. 3a). This is also visible on the $k_z$ dispersion for $k_z = 0.5$ r.l.u. (Fig. 3d, e). With $p$ polarization (Fig. 3d), the experimental periodicity of the band structure looks like it is doubled in comparison to the calculations due to these matrix elements effects. In fact, we recover the correct periodicity using $s$ polarization of the light (Fig. 3e). Note that the data of Fig. 3d cut the $M$ point in the $k_x - k_y$ plane between the first and second BZ, while the data of Fig. 3e cut the $M$ point between the second and third BZ. However, Fig. 3a–c indicate that variations of the photoemission matrix elements across different BZs are smaller than variations due to light polarization.

**Electronic joint density of states from density functional theory.** The SX-ARPES study detailed above shows that the band structure of $YNi_2B_2C$ is well described by our DFT calculations. Therefore, we rely on the analysis of the calculated FS (see Fig. S4) to assess the nesting properties of $YNi_2B_2C$ via the calculated eJDOS. The eJDOS along the [100] and [110] directions in $\mathbf{q}$ (Figs. 2c and 4a) lacks a peak at the position of the most pronounced $\mathbf{q} \approx X/2$ phonon broadening. However, our calculations reveal a peculiar momentum dependence when we look at the joint electronic density of states analyzed for fixed values of $k_z$, which we call a 2D-eJDOS. Detailed $k_z$-dependent calculations for phonon wave vectors $\mathbf{q}$ in the entire $[h, k, 0]$ plane reveal that the

2D-eJDOS is low for $k_z = 0$ (Fig. 4b) but increases strongly to $k_z = 0.5$ r.l.u. (Figs. 4c and S5 and S6).

We report the $k_z$ dependence of the 2D-eJDOS in more detail for three $\mathbf{q}$ ranges defined in Fig. 4b: (1) $\mathbf{q} = (h, k, 0)$ with $0 \leq h, k \leq 1$ [large blue square], (2) $\mathbf{q} \approx X/2$ [red bar] and (3) $\mathbf{q} \approx M$ [green bar]. Starting with range (1), the corresponding $\mathbf{q}$-averaged 2D-eJDOS, $\overline{2D - eJDOS}_q$, has a clear peak at $k_z = 0.5$ r.l.u. (Fig. 4d). Corresponding data for ranges (2) $\mathbf{q} = X/2$ (Fig. 4e) and (3) $\mathbf{q} = M$ (Fig. 4f) show that the peak of the $\overline{2D - eJDOS}_q$ at $k_z = 0.5$ r.l.u. is even more pronounced at the wave vectors of the strong phonon broadening. Overall, our analysis demonstrates that the 2D-eJDOS is extremely sensitive to $k_z$ and depends less on the phonon wave vector $\mathbf{q}$.

**Momentum-dependent phonon broadening from inelastic neutron scattering.** So far, we focused on the mechanism explaining the strong broadening of the two TA modes at $\mathbf{q} \approx X/2$ and $\mathbf{q} = M$ (Fig. 2a, b). While these effects are not sharply localized in $\mathbf{q}$ space, they do depend on the phonon wave vector[34] and only a rigorous experimental test enables an overall verification of the validity of the DFT calculations. Therefore, we performed neutron scattering experiments with improved energy resolution at phonon wave vectors along as well as off the high-symmetry directions investigated previously[30,32,34]. We compare our results to ab-initio calculations of $\gamma_{EPC}^q$ (see Eq. (1)), which were calculated using the linear response technique or density functional perturbation theory (DFPT)[48] in combination with the mixed-basis pseudopotential method[49].

For the present study, we measured phonons at wave vectors close to $\mathbf{q} = X/2$ and the $M$ point at room temperature and $T = 20$ K and looked for peak broadening. We observed broad but clear momentum dependences, e.g., in measurements going away from the $M$ point (see Fig. 2a, c, d): Already at close-by wavevectors, the broadening is clearly reduced (Fig. 2c) and completely absent further away in momentum space (Fig. 2d). More INS data taken along different directions for both anomalous modes are shown in Fig. S7 of SI. Figure 5 shows a comparison of the observed (symbols) and calculated (color code) phonon broadening in absolute wave vectors $\mathbf{Q} = \tau + \mathbf{q}$ to highlight the clear momentum dependences in 3D (see also Fig. S8 and Supplementary Note 2). Overall, we find good agreement.

Finally, we note that the $\mathbf{k}$-averaged eJDOS is about 50% higher for the $M$ point than for $\mathbf{q} \approx X/2$ (Figs. 2e and 4a), whereas the $\overline{2D - eJDOS}_q$ ($k_z = 0.5$ r.l.u.) (Figs. 4c, e, f and S6) and $\gamma_{EPC}^q$ (Fig. 5b, c) have similar values at these two wave vectors. This is further evidence that the electronic states at the Fermi level with $k_z = 0.5$ r.l.u. are selected by EPC matrix elements and mediate the coupling responsible for the broadening of the TA phonons in $YNi_2B_2C$.

## Discussion

EPC has been studied for a long time because it can stabilize emergent ground states as well as lead to phase competition. The particular role/impact of EPC matrix elements was discussed already in the 1970s[50,51], though experimental evidence for the decisive role of the momentum dependence of the EPC matrix elements was only reported in the last decade[24,26,27,52–54]. Still, only the $\mathbf{q}$ dependence of the EPC matrix element $g_{\vec{k}+\vec{q},\vec{k}}^{\vec{q}\lambda}$ was scrutinized.

Our work takes the full $\mathbf{k}$ and $\mathbf{q}$ momentum dependence of $g_{\vec{k}+\vec{q},\vec{k}}^{\vec{q}\lambda}$ into account. We can explain not only phonon broadening due to EPC as a function of $\mathbf{q}$ but also its pronounced weakening at increased temperatures in the absence of FS nesting and

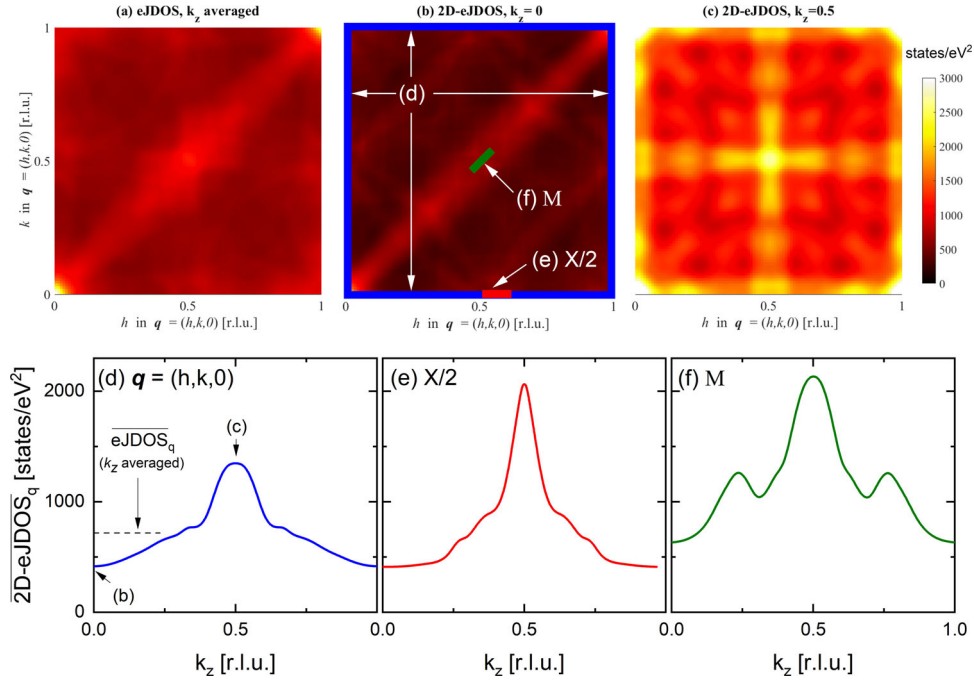

**Fig. 4 $k_z$-dependence of electronic joint density-of-states (eJDOS). a** eJDOS in the $\mathbf{q} = (h, k, 0)$ plane with $0 \leq h, k \leq 1$ r.l.u. averaged over the full 3D FS. **b, c** 2D-eJDOS($k_z$) considering only the FS states with (**b**) $k_z = 0$ and (**c**) $k_z = 0.5$ r.l.u.. All results are given in states/eV² and the same color-code applies to (**a–c**). **d–f** Detailed $k_z$ dependences of the $\mathbf{q}$-averaged 2D electronic joint density of states $\overline{2D - eJDOS_q}$ averaged over three different ranges in $\mathbf{q}$: (**d**) full $\mathbf{q} = (h, k, 0)$ plane [large blue square in (**b**)], (**e**) $\mathbf{q} \approx X/2$ [red bar in (**b**)] and (**f**) $\mathbf{q} = M$ [green bar in (**b**)], respectively. The black dashed horizontal line in (**d**) denotes the $k_z$-averaged value of $\overline{eJDOS_q}$ deduced from the data shown in (**a**).

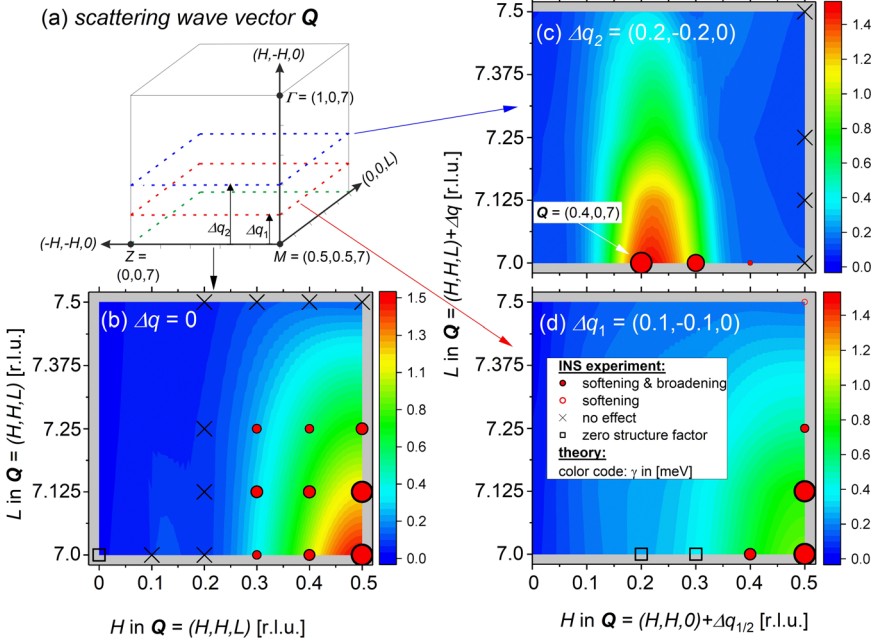

**Fig. 5 Momentum-dependent phonon broadening (DFPT + INS). a** Illustration of the scattering geometry in the horizontal [110]−[001] scattering plane. Measurements were also performed at out-of-plane wave vectors offset along the vertical $[1, \bar{1}, 0]$ direction by $\Delta\mathbf{q}_1 = (0.1, -0.1, 0)$ (red dashed plane) and $\Delta\mathbf{q}_2 = (0.2, -0.2, 0)$ (blue dashed plane). The $\Gamma$, $Z$ and $M$ points of the Brillouin zone are indicated (dots). **b–d** Color-coded plot of the calculated electronic contribution to the phonon line width $\gamma$ (see color bars for corresponding values in units of [meV]) for the transverse acoustic phonon mode investigated close to $\mathbf{Q} = (0.5, 0.5, 7) + \Delta\mathbf{q}_{1/2}$. Results are shown in the (H,H,L) scattering plane for (**b**) $\Delta\mathbf{q} = 0$, (**c**) $\Delta\mathbf{q}_1 = (0.1, -0.1, 0)$, and (**d**) $\Delta\mathbf{q}_2 = (0.2, -0.2, 0)$. Symbols refer to results from INS measurements and indicate phonon renormalization as indicated in the legend of panel (**d**). □ (*zero structure factor*) indicates that the phonon intensity was too weak to be observed—in agreement with DFPT structure factor calculations. Size of symbols (dots, circles) scale with the strength of the phonon renormalization within each panel. The white arrow in (**c**) indicates $\mathbf{Q} = (0.4, 0, 7)$ corresponding to a reduced wave vector $\mathbf{q} = (0.6, 0, 0)$.

anharmonicity. The peak of the $\overline{2D - eJDOS}_q$ as function of $k_z$ (Fig. 4d–f) accounts for a high sensitivity to temperature via the electronic band structure if the contribution of the electronic states with $k_z = 0.5$ r.l.u. to EPC is boosted by **k**-selective matrix elements. The peak of the $\overline{2D - eJDOS}_q(k_z)$ (Fig. 4d–f) is superficially reminiscent of classic FS nesting scenario where the peak in the **k**-averaged eJDOS(**q**) associated with the nesting condition is responsible for a phonon anomaly. At increased temperatures, smearing of the Fermi function quickly suppresses this peak in the eJDOS(q) and reduces the phonon broadening[18]. In our case, the peaks in the $\overline{2D - eJDOS}_q(k_z)$ (Fig. 4d–f) are expected to also be strongly broadened and reduced in height with increasing temperatures. This is exactly what is observed in experiments. Our results demonstrate that **k** selectivity of the EPC can boost the contribution of only a small part of the Fermi surface to the phonon lifetime. This new point of view has to be taken into account generally when assessing phonon anomalies in metallic systems.

This finding makes it necessary to reconsider our understanding of some well-known materials. NbSe$_2$ is a model system for various fundamental aspects (phase competition as function of dimensionality[12]; quantum phase transitions[55,56]; 2D quantum metal state[57]; Higgs mode in condensed matter physics[58,59]). The correct view of EPC is the indispensable ingredient for explaining many of the observed effects. In particular, NbSe$_2$ is considered to be a showcase for **q**-dependent EPC matrix elements featuring concomitant charge-density-wave (CDW) order ($T_{CDW} = 33$ K) and phonon-mediated superconductivity ($T_c = 7.2$ K)[60]. Calculations[25] and experiments[26] have shown that FS nesting is absent and concluded that the periodicity of the CDW is determined by the momentum dependence of the EPC matrix elements. Thus, the phonon broadening and softening close to $T_{CDW}$ is due to the EPC. However, lattice fluctuations overwhelm EPC away from $T_{CDW}$ and explain the strongly reduced phonon renormalization at elevated temperatures[24,25]. We find that the calculated eJDOS across the CDW ordering wavevector $\mathbf{q}_{CDW}$ in 2H-NbSe$_2$ is nearly featureless whereas the calculated line width due to EPC, $\gamma^q_{EPC}$, shows a pronounced broad maximum around $\mathbf{q}_{CDW}$ in agreement with experiment[27]. The situation is, in fact, very similar to that observed for the $X/2$ anomaly in YNi$_2$B$_2$C (see Figs. 2e and S8a). In analogy to the results presented here, the strong temperature dependence of the phonon line width in 2H-NbSe$_2$ could also originate from an interplay of **k**-selective EPC and the electronic band structure. In fact, Flicker et al.[25] already considered an orbital-dependent EPC matrix element. Yet, it remains unclear whether the model calculations also support a strongly **k** dependent 2D-eJDOS.

Recently, phonon anomalies related to CDW order in cuprates have attracted large scientific interest. In particular, a strong phonon broadening was observed at and below the onset of CDW fluctuations competing with superconductivity[4,6,8]. The abrupt decrease of the phonon line width in YBa$_2$Cu$_3$O$_{6.6}$ on entering the superconducting state[4] indicates a high-sensitivity to the opening of the superconducting energy gap on the FS mediated by the EPC. Yet, the mechanism determining the periodicity of the CDW remains unclear. The reported sharp **q** dependence of the phonon broadening is reminiscent of a FS nesting-type origin but the nesting could not be identified in the electronic band structure. **k** selective EPC in concert with large 2D-eJDOS on a small part of the FS is a novel approach for an improved understanding of these observations.

## Methods

**Inelastic neutron scattering**. Experiments were performed at the 1T and IN8[61] triple-axis spectrometers located at Laboratoire Léon Brillouin (LLB), CEA Saclay, and Institute Laue-Langevin (ILL), Grenoble, respectively. Double focusing pyrolytic graphite monochromators and analyzers were employed on the 1T

spectrometer. We used double-focusing copper (Cu200) monochromators and analyzers on IN8. A fixed analyzer energy of 14.7 meV allowed us to use a graphite filter in the scattered beam to suppress higher orders. The phonon scattering wave vector $\mathbf{Q} = \boldsymbol{\tau} + \mathbf{q}$ is expressed in reciprocal lattice units (r.l.u.) $\left(\frac{2\pi}{a}, \frac{2\pi}{b}, 2\pi/c\right)$ with the lattice constants $a = b = 3.51$ and $c = 10.53$ of the tetragonal unit cell. The single-crystal sample was mounted in a standard orange cryostat at ILL and in a closed-cycle refrigerator at LLB, allowing measurements down to $T = 2$ K.

**Soft x-ray angle-resolved photoemission spectroscopy**. experiments were performed on in-situ cleaved single crystals of YNi$_2$B$_2$C (001 surface) at the SX-ARPES endstation[62] at ADRESS beamline[63] of the Swiss Light Source (SLS) with incident photon energies $h\nu$ in the range $680-900$ eV (for more details see[62]). The increase of the photoelectron mean-free path related to the increased kinetic energy in the soft x-ray regime translates into a proportional improvement of the intrinsic $k_z$ resolution of the photoemission experiment, which is indispensable for our investigation. The measurements were performed at low sample temperature $T = 20$K in order to suppress relaxation of **k**-selectivity because of thermal motion[64]. For consistency with the results from neutron scattering, the electron scattering wave vector **k** is expressed in r.l.u. $\left(\frac{2\pi}{a}, \frac{2\pi}{b}, 2\pi/c\right)$.

**Density functional theory**. calculations were carried out in the framework of the mixed basis pseudopotential method[47] using the local density approximation (LDA) (for details see ref.[34]) with a $80 \times 80 \times 30$ grid of points in **k** space. Plots of the electronic band structure and its properties are based on an interpolated $200 \times 200 \times 30$ grid to enhance the in-plane accuracy. All results presented here were obtained for the experimental lattice constants of the tetragonal structure ($a = b = 3.51$, $= 10.53$) with internally relaxed atomic coordinates. Phonon properties and EPC matrices were calculated for a $32 \times 32 \times 16$ grid in phonon, i.e. **q** momentum space using the linear response technique or density functional perturbation method (DFPT)[48] in combination with the mixed-basis pseudopotential method[49]. While the eJDOS is defined as an integral over the 3D Brillouin zone (see main text), the 2D-eJDOS is evaluated for a fixed value of $k_z$, i.e., it samples the electronic phase-space over slices parallel to the $k_x - k_y$ plane. δ-functions are approximated by Gaussians of width $\sigma = 0.05$ eV.

## Data and code availability

Data used for figures are available at ref.[65]. More detailed information including details of the ab-initio calculations are available from the corresponding author upon request.

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

## Acknowledgements

We acknowledge local support from Daria Sostina during test experiments at the PEARL beamline at SLS. This project was supported from the Swiss National Science Foundation (SNSF) Grant No. P00P2_170597. We acknowledge the Paul Scherrer Institute, Villigen, Switzerland for provision of synchrotron radiation beamtime at beamline ADRESS of the Swiss Light Source. D.R. was supported by the DOE, Office of Basic Energy Sciences, Office of Science, under Contract No. DE-SC0006939.

## Author contributions

Neutron scattering: P.K., J.-P.C., A.I., D.R., F.W.; photo-emission spectroscopy: G.K., C.N., P.K., M.R., B.S., T.J., V.S., M.M., P.N., C.M., F.W.; theory: R.H., P.K., F.W.; data analysis: P.K., G.K., C.M., F.W.; manuscript: G.K., C.N., T.J., D.R., C.M., F.W.; F.W. coordinated the project.

## Funding

## Competing interests

The authors declare no competing interests.
