## [Peer Review File · Nature Communications]

REVIEWER COMMENTS

Reviewer #1 (Remarks to the Author):

This manuscript focuses on the electron phonon coupling (EPC) in YNi₂B₂C that the EPC is found to be strong at X/2 and M point. The authors claim that Fermi nesting and phonon anharmonicity are not important for the phonon broadening, based on scattering experiments and first-principles calculations. The experimental and theoretical finding of strong EPC at X/2 and M points are solid and plausible; however, overall speaking this study is rather technical, which I think is not suitable for publication in journals of broad interests such as Nature Communications. It does not offer a comprehensive understanding about EPC in this system except the probed space and phonon mode, which is still limited in the BZ. What it lacks more is actually a link of this specific type of EPC to the physical properties that the readers may be interested in. Despite the discussions about CDW and superconductivity, I feel the connection is unclear or lacking evidence.

Some other comments:

1- The authors mention that the EPC matrix "is expected to be temperature-independent for temperatures of the order of up to 1000 K". This is incorrect, since both the phonon and electronic band structures should have temperature dependence.

2- The strongest phonon anomalies are found at two q points, i.e., (0.5,0.5,0) and (0.55,0,0), which are denoted as M and X/2. I am wondering why X point is located at (1,0,0) of the BZ, instead of (0.5,0,0)? Usually (0.5,0,0) is the X point.

3- To get a more complete understanding of EPC, I would suggest the authors to calculate the phonon linewidths for all the high-symmetry q lines and all phonon branches via DFPT. This can tell if the M and X/2 points are the only two places where strong EPC occurs.

4- Details regarding how the 2D-eJDOS are calculated are not provided in the Methods section or main text.

Reviewer #2 (Remarks to the Author):

The work reported in the manuscript focuses on understanding the microscopic origin of strong phonon damping (increased linewidths) observed for certain wavevectors in the normal state of the electron-phonon superconductor YNi₂B₂C at low temperature.

Authors argue that previously proposed explanations for the phonon broadenings in YNi₂B₂C in terms of FS nesting do not hold. They corroborate this by comparing their soft x-ray angle-resolved photoemission spectroscopy (SX-ARPES) measurements with an ab-initio lattice dynamical and electronic band structure calculations. What authors find is that not only the calculations account reasonably well for the observed electronic spectra, but also that the calculated (using the DFT perturbation theory) phonon damping agrees well with the inelastic neutron scattering (INS) measurements of phonons. On the other hand, the FS nesting effects, while important for explaining the temperature dependence, do not have enough spread in wave vector to explain the observed linewidth and its variation with q . The main point that authors make is that the matrix element of electron-phonon coupling (EPC) and its wave vector dependence are important and need to be adequately accounted for in order to understand the phonons in YNi₂B₂C and by extension in other materials of interest. I think this is an interesting and important result, which might have significant impact in the field. The wave vector dependence of the EPC is usually considered unimportant and often neglected. The actual matrix element of EPC is thus replaced with a constant, such as in the standard BCS theory of superconductivity. Clear experimental demonstrations of the inadequacy of this approximation in an electron-phonon superconductor are important but rare. The present work does furnish such a convincing demonstration. Overall, the experimental results are of high quality and theoretical work is also commendable. I recommend that the manuscript is accepted for publication in Nature Communications.

I only have a couple of comments for authors to consider when preparing the revised manuscript.

1. I find that Fig. S8 in the Supplementary is much more revealing and convincing than the corresponding Fig. 5 in the main text. I would suggest moving Fig. S8 in place of Fig. 5 of the main manuscript while moving the line scans shown in Fig. 5 to Fig. 1, combining with similar scans shown in Fig. 1. The manuscript would need to be revised correspondingly.

2. In the discussion of lattice dynamics in the background section, lattice symmetry and parameters should be specified before referring to special points of the Brillouin zone (BZ).

Reviewer #3 (Remarks to the Author):

Interesting work by Kurzhals and coworkers combining measurements and calculations to study electron-phonon coupling (EPC) behavior and associated characteristics in the well-known conventional superconductor YNi₂B₂C. The authors used ARPES and INS, underpinned by DFT-based calculations in a convincing and robust way. The strong point here is the extension of the study to probe the effect of specific values of electron-momentum on the strength of EPC. The paper is well written and deserves publication in Nature Communications as it is, given the potential insight it seems providing into a further understanding of - at least - the conventional superconducting phase.

MAIN EDITORIAL CHANGES:

Figures:

Fig. 2:

We include here the raw data in panels (c,d) previously shown in Figure 5(a,b).

Fig. 3:

In panel (f) we show an extended scheme of Brillouin zones to better visualize the momentum positions of the discussed anomalies.

Fig. 5:

Here we show the previous figure S8, which demonstrates the 3D momentum dependence of electron-phonon coupling very well.

Text:

- Page 1, 3rd paragraph:

The sentence

“Yet, $g_{\vec{k}+\vec{q},\vec{k}}^{\vec{q}\lambda}$ is expected to be temperature-independent for temperatures of the order of up to 10^3 K”

was changed to

“Yet, changes of the phonon line width because of the temperature dependence of $g_{\vec{k}+\vec{q},\vec{k}}^{\vec{q}\lambda}$ up to room temperature are expected to be small.”

- Page 3, 3rd paragraph:

Information about the structure of $\text{YNi}_2\text{B}_2\text{C}$ is provided.

- Page 7, 1st paragraph

We revised the text referring to the revised Figure 5.

- Page 9, 1st paragraph (Materials and Methods)

We added text regarding the way the 2D-eJDOS is calculated.

Reviewer #1 (Remarks to the Author):

This manuscript focuses on the electron phonon coupling (EPC) in YNi₂B₂C that the EPC is found to be strong at X/2 and M point. The authors claim that Fermi nesting and phonon anharmonicity are not important for the phonon broadening, based on scattering experiments and first-principles calculations. The experimental and theoretical finding of strong EPC at X/2 and M points are solid and plausible; however, overall speaking this study is rather technical, which I think is not suitable for publication in journals of broad interests such as Nature Communications. It does not offer a comprehensive understanding about EPC in this system except the probed space and phonon mode, which is still limited in the BZ. What it lacks more is actually a link of this specific type of EPC to the physical properties that the readers may be interested in. Despite the discussions about CDW and superconductivity, I feel the connection is unclear or lacking evidence.

We regret that the referee does not see the impact of our result for the research on systems involving electron-phonon coupling. Contrary to the referee's statement, our model offers a comprehensive understanding of the electron-phonon coupling in this system [see response to (3) below]. This may be better demonstrated by the new Figure 5 (former Figure S8, following request of referee #2).

The effect we demonstrate for YNi₂B₂C, i.e., electron-momentum dependence of electron-phonon coupling can lead to similar effects as Fermi surface nesting, was overlooked so far in the research on corresponding electronic phase transitions. Therefore, it needs to be taken into account considering well-studied seminal compounds as well as emergent materials.

In the discussion section, we have described two puzzling cases for which our work could have a significant impact. It is not the purpose of the current paper to give a final explanation for the origin of the CDW phases in these materials, but we outline a few similarities between the cases of YNi₂B₂C and of NbSe₂ and YBa₂Cu₃O_{6.6} that possibly point towards a similar mechanism.

Some other comments:

1- The authors mention that the EPC matrix "is expected to be temperature-independent for temperatures of the order of up to 1000 K". This is incorrect, since both the phonon and electronic band structures should have temperature dependence.

The referee is correct. We have revised the corresponding sentence in the introduction.

What we want to express is that large changes of the phonon linewidth – with which we are concerned in the current manuscript – are not likely to originate from a temperature dependence of the EPC matrix element. The matrix elements depend on both the electronic and phononic band structure and, in general, are sensitive to temperature dependent changes of these quantities. Yet in the absence of particular effects such as phase transitions or unusually strong lattice anharmonicity, electronic properties (disregarding the population of states close to the Fermi energy covered by the Fermi function) with dispersion in the range of [eV] change significantly only at correspondingly high temperatures. Lattice dynamical properties change at lower temperatures because of the much smaller excitation energies. However, in the temperature range up to room temperature and often even above, lattice dynamics - in the absence of particular effects such as phase transitions or

unusually strong lattice anharmonicity - are typically well understood by the effects of thermal expansion and do not significantly alter the EPC matrix elements.

We revise the criticized sentence along these lines and apologize for the inaccurate original statement.

2- The strongest phonon anomalies are found at two q points, i.e., (0.5,0.5,0) and (0.55,0,0), which are denoted as M and X/2. I am wondering why X point is located at (1,0,0) of the BZ, instead of (0.5,0,0)? Usually (0.5,0,0) is the X point.

In the structure of YNi₂B₂C, X points are equivalent to Z. This can be illustrated by the following consideration:

In the used setup of the unit cell, zone center are defined by: $h+k+l = \text{“even”}$

Zone centers: (0,1,23), (0,0,22)

Q = (0,0,23) (not a zone center)

- X point considering the Zone center (0,1,23)
- Z point considering the zone center (0,0,22)

This is illustrated in the following figure, which replaces the previous BZ scheme in Figure 3f:

3- To get a more complete understanding of EPC, I would suggest the authors to calculate the phonon linewidths for all the high-symmetry q lines and all phonon branches via DFPT. This can tell if the M and X/2 points are the only two places where strong EPC occurs.

We did this in previous work published in References 32 and 33. In particular the latter reference shows in detail the information requested by the referee and highlights the strong phonon anomalies at the M and X/2 points discussed in the current manuscript.

Phonon anomalies are not restricted to these two points but the EPC strength is the strongest. Figs. 5, S8 and S9 of the initially submitted manuscript show the situation for the low energy phonons.

We critically re-read the text but believe that these facts are clearly stated several times, e.g.,

- Introduction: **line 62**
- lattice dynamics in YNi₂B₂C: **lines 113-121**
- Momentum-dependent phonon broadening from inelastic neutron scattering: **lines 205-212**

4- Details regarding how the 2D-eJDOS are calculated are not provided in the Methods section or main text.

We provide an explanation on the 2D-eJDOS in the revised version of the manuscript in the Methods section.

Reviewer #2 (Remarks to the Author):

The work reported in the manuscript focuses on understanding the microscopic origin of strong phonon damping (increased linewidths) observed for certain wavevectors in the normal state of the electron-phonon superconductor YNi₂B₂C at low temperature.

Authors argue that previously proposed explanations for the phonon broadenings in YNi₂B₂C in terms of FS nesting do not hold. They corroborate this by comparing their soft x-ray angle-resolved photoemission spectroscopy (SX-ARPES) measurements with an ab-initio lattice dynamical and electronic band structure calculations. What authors find is that not only the calculations account reasonably well for the observed electronic spectra, but also that the calculated (using the DFT perturbation theory) phonon damping agrees well with the inelastic neutron scattering (INS) measurements of phonons. On the other hand, the FS nesting effects, while important for explaining the temperature dependence, do not have enough spread in wave vector to explain the observed linewidth and its variation with q . The main point that authors make is that the matrix element of electron-phonon coupling (EPC) and its wave vector dependence are important and need to be adequately accounted for in order to understand the phonons in YNi₂B₂C and by extension in other materials of interest.

I think this is an interesting and important result, which might have significant impact in the field. The wave vector dependence of the EPC is usually considered unimportant and often neglected. The actual matrix element of EPC is thus replaced with a constant, such as in the standard BCS theory of superconductivity. Clear experimental demonstrations of the inadequacy of this approximation in an electron-phonon superconductor are important but rare. The present work does furnish such a convincing demonstration. Overall, the experimental results are of high quality and theoretical work is also commendable. I recommend that the manuscript is accepted for publication in Nature Communications.

We thank the referee for the positive assessment of our work with regard to its quality as well as significance for the field of electron-phonon coupling.

I only have a couple of comments for authors to consider when preparing the revised manuscript.

1. I find that Fig. S8 in the Supplementary is much more revealing and convincing than the corresponding Fig. 5 in the main text. I would suggest moving Fig. S8 in place of Fig. 5 of the main manuscript while moving the line scans shown in Fig. 5 to Fig. 1, combining with similar scans shown in Fig. 1. The manuscript would need to be revised correspondingly.

The revised manuscript includes the suggested change in the figures and the text was revised accordingly.

2. In the discussion of lattice dynamics in the background section, lattice symmetry and parameters should be specified before referring to special points of the Brillouin zone (BZ).

The information is now given at the suggested position in the text.

Reviewer #3 (Remarks to the Author):

Interesting work by Kurzhals and coworkers combining measurements and calculations to study electron-phonon coupling (EPC) behavior and associated characteristics in the well-known conventional superconductor YNi₂B₂C. The authors used ARPES and INS, underpinned by DFT-based calculations in a convincing and robust way. The strong point here is the extension of the study to probe the effect of specific values of electron-momentum on the strength of EPC. The paper is well written and deserves publication in Nature Communications as it is, given the potential insight it seems providing into a further understanding of - at least - the conventional superconducting phase.

We also thank referee #3 for the fully positive response.

REVIEWERS' COMMENTS

Reviewer #1 (Remarks to the Author):

The authors have adequately addressed my technical questions. However, although the finding of enhanced EPC at specific wave vectors is certainly interesting, it is not unexpected as it is the key ingredient of the phonon linewidth or broadening. I think this is a solid study, but it is more suitable for a specialized journal. As mentioned in my first report, it is hard for this work to attract broad interests.

Reviewer #2 (Remarks to the Author):

In my opinion, in the revised manuscript authors have satisfactorily addressed all Reviewer comments. The revision have also improved the overall quality of the manuscript. I recommend that the manuscript is accepted for publication in Nature Communications.